# Enhanced Skin Permeation of 5-Fluorouracil through Drug-in-Adhesive Topical Patches

**DOI:** 10.3390/pharmaceutics16030379

**Published:** 2024-03-10

**Authors:** Sangseo Kim, Souha H. Youssef, Kyung Min Kirsten Lee, Yunmei Song, Sachin Vaidya, Sanjay Garg

**Affiliations:** 1Centre for Pharmaceutical Innovation, Clinical and Health Sciences, University of South Australia, Adelaide, SA 5000, Australia; sangseo.kim@mymail.unisa.edu.au (S.K.); souha.youssef@mymail.unisa.edu.au (S.H.Y.); may.song@unisa.edu.au (Y.S.); 2Central Adelaide Local Health Network, The Queen Elizabeth Hospital, Woodville, SA 5011, Australia; vaidya.2002@gmail.com

**Keywords:** 5-fluorouracil, Eudragit^®^ E, chemical permeation enhancers, Strat-M^®^ membrane, drug-in-adhesive patches, non-melanoma skin cancer

## Abstract

5-fluorouracil (5-FU), commercially available as a topical product, is approved for non-melanoma skin cancer (NMSC) treatment with several clinical limitations. This work aimed to develop 5-FU-loaded topical patches as a potential alternative to overcome such drawbacks. The patches offer accurate dosing, controlled drug release and improved patient compliance. Our study highlights the development of Eudragit^®^ E (EuE)-based drug-in-adhesive (DIA) patches containing a clinically significant high level of 5-FU (approximately 450 µg/cm^2^) formulated with various chemical permeation enhancers. The patches containing Transcutol^®^ (Patch-TRAN) or oleic acid (Patch-OA) demonstrated significantly higher skin penetration ex vivo than their control counterpart, reaching 5-FU concentrations of 76.39 ± 27.7 µg/cm^2^ and 82.56 ± 8.2 µg/cm^2^, respectively. Furthermore, the findings from in vitro permeation studies also validated the superior skin permeation of 5-FU achieved by Patch-OA and Patch-TRAN over 72 h. Moreover, the EuE-based DIA patch platform demonstrated suitable adhesive and mechanical properties with an excellent safety profile evaluated through an inaugural in vivo human study involving 11 healthy volunteers. In conclusion, the DIA patches could be a novel alternative option for NMSC as the patches effectively deliver 5-FU into the dermis layer and receptor compartment ex vivo for an extended period with excellent mechanical and safety profiles.

## 1. Introduction

5-Fluorouracil (5-FU) is a pyrimidine antimetabolite chemotherapy drug commonly used to treat different cancers, including non-melanoma skin cancer (NMSC). It interferes with the normal metabolic processes of cancer cells to exhibit its anticancer activities, specifically by inhibiting the activity of thymidylate synthase, which is involved in synthesising the building blocks of DNA [1].

5-FU is commercially available as a topical cream (e.g., Efudix^®^, Efudex^®^, Carac^®^, among others) for the treatment of superficial basal cell carcinoma (BCC), squamous cell carcinoma in situ (Bowen’s disease) and other precancerous skin lesions including actinic keratosis (AK) [2]. However, there are some clinical limitations to the use of 5-FU topical cream for NMSC treatment. These include poor drug permeability into the deeper skin layers and limited efficacy against more aggressive skin cancer types such as nodular BCC and melanoma. In addition, a lack of consensus on clinically relevant topical doses resulted in its application either excessively or sub-optimally, with multiple daily applications and a prolonged treatment period, possibly leading to poor patient compliance [3].

Topical patches offer several clinical advantages including ease of administration and facilitation of the patient’s commitment to the treatment protocol, which can potentially improve the drawbacks associated with commercial creams [4,5]. The patches are also easily cut to any size and directly applied to the affected areas without the need for an applicator or protective equipment [3]. In addition, patches can generally deliver a specific dose over a prolonged period of 1 to 7 days, reducing dosing frequency significantly compared to cream formulations which often require multiple daily applications [6].

However, 5-FU needs to penetrate the hydrophobic physical barrier of the outermost skin layer called the Stratum corneum (SC) to exert its therapeutic actions at the tumour site. The physicochemical properties of an active ingredient critically affect its permeability through the SC. Some of the desired properties that favour skin permeation may include a high aqueous solubility (>1 mg/mL), a low molecular weight (<500 Da), and a low melting point (<200 °C) as well as a partitioning coefficient (Log P) of 1 to 3 [7]. Whilst 5-FU’s high melting point (291.82 °C) and low Log P (–0.85) may be indicative of unfavourable diffusion into the skin, an adequate aqueous solubility of 16.76 mg/mL and a low molecular weight of 130.01 g/mol may favour its skin permeation [8]. Although the accurate prediction of drug permeation using only a few physicochemical properties may not always be feasible, several studies demonstrated that 5-FU alone would generally have limited skin permeation [9,10].

Various techniques including nanotechnology [11], microneedles [12], and the use of chemical permeation enhancers (CPEs) [13,14,15] have been explored to enhance skin permeation for many drugs. CPEs have been adopted extensively due to their benefits including versatility, good safety profile, ease of formulation, cost-effectiveness, and simplified scale-up processes, encouraging their widespread application in topical drug delivery [16]. CPEs may be classified based on their chemical structures as hydrocarbons, alcohols, acids, amines, amides, esters, carboxylic acids, and terpenes, among others [17]. Previous studies have demonstrated the effects of several CPEs for the delivery of 5-FU into the excised human skin [10,18]. In the study, the pre-treatment of the skin samples with oleic acid (OA), menthone, 1,8-cineole, or nerolidol achieved permeation enhancement of 5-FU solution up to 24-, 42-, 95- and 25-fold after 12 h, respectively [10]. Similarly, another study revealed that aqueous solutions containing 5-FU with 5% (*w*/*v*) isopropyl myristate, 3% (*w*/*v*) Azone^®^, or 5% (*w*/*v*) lauryl alcohol improved the 5-FU permeation up to 3-, 24- and 4-fold through excised human SC [18].

Adhesion is one of the critical quality attributes of topical patches [3]. It ensures the entire surface of a patch maintains full contact with the skin for an intended application period, facilitating the transfer of drug molecules to the target site [3,19]. Our previous study [3] reported the development of DIA topical patches containing 5-FU for the first time using Eudragit^®^ E100 (EuE) as a matrix-forming polymer. EuE is a cationic copolymer consisting of dimethylaminoethyl methacrylate, butyl methacrylate, and methyl methacrylate at the ratio of 2:1:1 and can produce flexible films suitable for topical applications when formulated with an appropriate type and amount of plasticiser. In our study, various types of plasticisers and their concentrations were screened as EuE alone could not produce patches with suitable mechanical and adhesive properties for topical use. In addition, the physicochemical compatibility of the excipients and 5-FU was fully evaluated, and their composition was optimised to develop the 5-FU DIA patches with optimal mechanical and adhesive properties for adequate wear comfort and adhesion for the intended duration of the application. Moreover, the in vitro release studies demonstrated that 5-FU was released from EuE-based patches in a controlled manner suitable for topical application.

In this study, we aimed to incorporate a high dose of 5-FU at a clinically significant level relevant to the topical treatment of NMSC. This study also highlights the ex vivo and in vitro deposition and permeation performance of the 5-FU-loaded patches for the first time formulated with various chemical permeation enhancers when they aim to deliver their loads over 72 h. Lastly, our study was the first to conduct the in vivo human assessment of EuE-based DIA patches with a specific focus on their adhesive and sensory properties, as well as their safety profiles. Efudex^®^ cream has also been used for comparison purposes.

## 2. Materials and Methods

### 2.1. Materials

5-FU (>99%) was obtained from Hangzhou Dayang Chem Co. Ltd. (Hangzhou, Zhejiang, China). Evonik Industries AG (Essen, North Rhine-Westphalia, Germany) provided EuE as a gift. Scotchpak™ 9709 and Cotran™ 9720 were supplied as a gift by 3M (Maplewood, MN, USA). Transcutol^®^ HP was gifted from Gattefosse (Saint-Priest, France). Succinic acid (BioXtra, ≥99.0%), sodium azide, sodium dodecyl sulfate (SDS), triacetin (99%), and Tween^®^ 20 were purchased from Sigma-Aldrich (Castle Hill, NSW, Australia). OA was obtained from PCCA (Houston, TX, USA). Efudix^®^ cream was obtained from Australian Pharmaceutical Industries (Wingfield, SA, Australia). Strat-M^®^ membranes (25 mm) were obtained from Merck Millipore (Macquarie Park, NSW, Australia). D-Squame^®^ pressure instrument (D500) and D-Squame^®^ standard sampling discs (D100) were obtained from Clinical & Derm (Dallas, TX, USA). All other reagents and solvents used throughout the experiments were of analytical or HPLC grade and used without further modification.

### 2.2. Slide Crystallisation Studies

The maximum solubility of 5-FU in the EuE copolymer plasticised with triacetin was determined by slide crystallisation studies with slight modifications [20]. Briefly, EuE solutions plasticised with 40% (to polymer ratio, *w*/*w*) triacetin containing various concentrations of 5-FU (3%, 5%, 6%, 7%, and 10% drug-to-polymer ratios, *w*/*w*) were prepared. A few drops of the mixture were placed on a glass slide and allowed to dry at room temperature. An optical light microscope (Olympus BX51, Tokyo, Japan) was used to observe the presence of crystals. A 5-FU solution in methanol (MeOH) was also used for comparison.

### 2.3. Preparation of 5-FU-Loaded Drug-in-Adhesive Patches

5-FU-loaded adhesive mixtures were prepared as per a previous method with slight modifications (Table 1) [3]. Briefly, 30 mL of MeOH was added to 5-FU, EuE and succinic acid to fully dissolve the ingredients. The mixture was then heated to 70 °C under continuous stirring to evaporate excess MeOH to achieve the approximate mixture weight of 15 g. After adding triacetin and chemical permeation enhancers, the mixture was made up to the final casting solution weight of 20 g with MeOH. The bottle was tightly sealed, and the mixture was mixed until a uniform solution was formed. Using a film applicator, the casting solution was then cast on Scotchpak™ 9709 (release liner) and left at room temperature to dry for 24 h. The Cotran™ 9720 (backing layer) was gently attached to the DIA film using a hand roller and the prepared 5-FU DIA patches were stored in the dark in an air-tight container until further use.

### 2.4. HPLC Method for Quantification of 5-FU

The determination of 5-FU contents was performed using an HPLC method as previously described [21]. A 250 × 4.6 mm, 5 µm (PhenoSphere-Next™ C18) column was used on a Shimadzu LC system (Shimadzu Corporation, Kyoto, Japan). The HPLC machine was equipped with an autosampler (SIL-20A HT), a degasser (DGU-20A3), and a pump (LC-20ADXR) for analysis. The composition of the mobile phase was MeOH/water (2:98 *v*/*v*, pH 2.5 adjusted with phosphoric acid) containing 0.5 mM SDS. Other parameters include the total HPLC runtime (12 min), the flow rate (0.5 mL/min), the injection volume (10 µL), and the UV detection wavelength (265 nm).

### 2.5. Characterisations of the Patches

#### 2.5.1. Content Uniformity

The patches were randomly cut into small pieces using a hole punch (9 mm). After fully dissolving each piece (*n* = 6) in 10 mL of MeOH for 24 h under continuous shaking, the solution was diluted with the mobile phase and filtered through a 0.45 µm acetate syringe filter. The samples were analysed according to the HPLC method for the quantification of 5-FU content.

#### 2.5.2. Thickness, Weight, Folding Endurance, and Surface pH

The overall thickness of the patches including a release liner, adhesive matrix, and backing layer was measured using a digital micrometre (IDS1012, Mitutoyo, Japan) (*n* = 6). In addition, the average patch weight was taken from six small pieces (9 mm) using a digital balance. The folding endurance was evaluated by folding the patches repeatedly at the same position until breakage or reaching 100 times. The surface pH of the patches was determined as previously described [3]. Briefly, each patch was placed in a separate test tube containing 1 mL of water and allowed to rest for one hour. A digital pH metre (Orion Star A121, Thermo Scientific, Waltham, MA, USA) probe was then placed near the patch surface to obtain the pH readings.

#### 2.5.3. Percentage Moisture Content

Small pieces (9 mm) were cut out from the patches and kept in a silica bead-containing desiccator at 25 °C. The weight of the patches was measured after 72 h using a digital balance and the percentage moisture content of each patch was determined according to the following Equation (1):(1)$$\mathrm{Moisture}\;\mathrm{content}\;(\%)=\frac{Initial\;weight-Final\;weight}{Initial\;weight}\times 100\;$$

#### 2.5.4. Percentage Moisture Absorption

Small pieces (9 mm) were cut out from the patches and kept in a desiccator filled with 200 mL of saturated potassium chloride solution at 25 °C to obtain a relative humidity (RH) of 84%. The weight of the patches was measured after 72 h with a digital balance. The percentage moisture absorption of each patch was calculated according to the below Equation (2):(2)$$\mathrm{Moisture}\;\mathrm{absorption}\;(\%)=\frac{Final\;weight-Initial\;weight}{Initial\;weight}\times 100$$

#### 2.5.5. Scanning Electron Microscopy Analysis

Scanning electron microscopy images of the pure 5-FU, surface morphology, and cross-section of the DIA patch were taken using a Zeiss Merlin Field-Emission Gun equipped with a silicon drift detector energy-disperse X-ray spectroscopy (Jena, Thuringia, Germany). The samples were fixed on the sample holder using double-sided tape. In the case of the patch sample to show its cross-section, a small thin piece was vertically positioned. The sample was sputter-coated with platinum and examined using the accelerating voltage of 2 kV.

#### 2.5.6. Probe Tack Test

The tack of the adhesive patches was measured using a TA.XTplus Texture Analyser (with a 5 kg load cell) (Stable Micro Systems, Godalming, Surrey, UK) as previously described [3]. Briefly, the patch was cut to a smaller size (20 mm × 20 mm) and the backing side of the patch was fixed on the surface of the texture analyser using a double-sided tape. The release liner was gently peeled from the patch to expose the adhesive side of the patch towards the 15 mm flat cylindrical stainless-steel probe. The probe was applied onto the adhesive surface at a consistent force (0.05 N) for 5 s and detached at the return speed (5 mm/s). The presence of adhesive residues on the probe surface was visually checked and the forces required for the detachment were recorded.

#### 2.5.7. The 90° Peel Adhesion Test

The 90° peel adhesion of the adhesive patches was measured using a TA.XTplus Texture Analyser (Stable Micro Systems, Godalming, Surrey, UK) equipped with adhesive indexing and a 90° peel rig accessory [3]. Briefly, a long strip (100 mm × 10 mm) was cut out of the patches and fixed to the stainless-steel test bed at a consistent force using a hand roller. One side of the strip was attached to a tensile grip of the machine, and it was removed from the panel at a 90° angle at a speed of 300 mm/min. The average force required to peel the strips was recorded for each formulation.

### 2.6. Ex Vivo Permeation and Deposition Studies

#### 2.6.1. Porcine Skin Preparation

Porcine ears were obtained from a local abattoir (Murray Bridge, SA, Australia). Before tissue collection, the Animal Ethics Committee at University of South Australia was notified by submitting a “notification of use of scavenged tissues”. After collection, the ears were gently washed with water and the dorsal side was excised from the cartilage using a scalpel. Subcutaneous tissues and excess hair were trimmed using a pair of scissors and an electric hair clipper. The skin samples were dermatomed (500 µm thickness) using a dermatome (75 mm, Zimmer Inc, Columbus, OH, USA), placed in an aluminium foil, and stored at −20 °C until use.

#### 2.6.2. Skin Integrity Evaluation

The skin integrity was evaluated before conducting each experiment. The skin samples were assessed based on visual examinations and transepithelial electrical resistance (TEER). Briefly, the damage-free skin samples with uniform thickness were placed on Franz diffusion cells filled with the release medium (0.1 M phosphate-buffered saline, pH 6.5) containing sodium azide (0.02% *w*/*v*) with the SC side facing upwards. After an equilibration period of 15 min, the TEER of the skin samples was determined using a digital multimeter (MM400, Klein Tools, Dandenong South, VIC, Australia). Only skin samples with TEER > 10 kΩ were used for the ex vivo studies [22].

#### 2.6.3. Ex Vivo Permeation Studies

Ex vivo permeation and deposition studies were conducted using Franz diffusion cells (application area of 0.64 cm^2^) as previously described [3]. Briefly, the receptor compartment (5.2 mL) was filled with the release medium (0.1 M phosphate-buffered saline, pH 6.5) containing sodium azide (0.02% *w*/*v*) under continuous stirring with a magnetic bar. The skin samples with good barrier integrity as previously determined were cut to a diameter of 25 mm using a hole punch and placed onto the Franz diffusion cells with the SC side facing towards the donor chamber. The 5-FU patches were cut into small pieces (9 mm diameter) using a hole punch and placed on the SC side of the skin. The temperature of Franz diffusion cells was controlled at 32 °C using a circulating water bath system. Samples (1000 µL) were taken at a pre-determined interval and replaced with an equal volume of the release medium to maintain the sink condition. The 5-FU contents in the samples were analysed according to the HPLC method. Efudix^®^ cream (5%) was also used for comparison purposes.

#### 2.6.4. Quantification of 5-FU Deposition in Skin Samples

The amounts of 5-FU deposited in different skin layers were determined as previously described with slight modifications [9]. Firstly, the dosed area on the pig skin was gently washed with MilliQ^®^ water (10 mL) to remove excess formulations on the surface and dried at room temperature for 2 h. The SC layer of the skin was then removed by a tape-stripping method. Briefly, D-Squame^®^ standard sampling discs were fixed to the skin sample with gentle pressure applied using a D-Squame^®^ pressure instrument and removed using forceps after 5 s. The process was repeated 15 times. Subsequently, the epidermis layer was separated gently from the rest of the skin sample using forceps and tweezers. No additional technique such as heat separation was required. Each skin sample was cut into smaller pieces and placed in 1 mL of MeOH for the extraction of 5-FU. After vortexing for 2 min, all samples were placed in an ultrasonic bath for 45 min and filtered through a 0.22 µm cellulose acetate syringe filter. The 5-FU contents in each skin sample were determined according to the HPLC method.

### 2.7. In Vitro Permeation Studies Using Strat-M^®^ Membranes

The in vitro permeation studies were performed according to the protocol used for ex vivo permeation, except the pig ear skin, which was replaced with Strat-M^®^ synthetic membranes. The samples (1 mL) were collected at a pre-determined interval of up to 72 h and analysed using the HPLC method.

### 2.8. Stability Studies

The 5-FU-loaded DIA patches were packed in an aluminium foil bag and stored at room temperature (25 ± 2 °C, 65% RH) for 6 months. The samples were visually assessed, and the 5-FU contents were analysed using the HPLC method. Changes in adhesive properties over the period were also evaluated, as described in Section 2.5.6.

### 2.9. In Vivo Evaluation of the Adhesive Properties and Safety of the Patches

An in vivo human study involving 11 healthy volunteers aged over 18 years old was conducted over 72 h to evaluate the adhesive properties, wear comfort, and potential safety of the drug-free DIA patches. All procedures were performed in compliance with relevant laws and institutional guidelines and were approved by the Human Research Ethics Committee (University of South Australia, Approval number: 205191). Informed consent was also obtained before the experiment. Each volunteer was advised not to use any moisturisers on their arms 12 h before the appointment. The volunteers were supplied with a set of drug-free counterparts of Patch-OA and Patch-TRAN and advised to apply them on each arm with gentle pressure for 30 s. They were also advised to avoid contact with excess water and soap during the study. Upon the completion of the 72 h study, each participant was asked to complete a questionnaire comprising six questions to evaluate various aspects of sensory and adhesive properties and potential local toxicities. A 5-point Likert scale was used for each question, except for listing side effects, and the scores were compared using Student’s *t*-test.

### 2.10. Statistical Analysis

All statistical analysis was performed using GraphPad Prism software (version 10.0.1, San Diego, CA, USA). A one-way analysis of variance (ANOVA) followed by Tukey’s multiple comparisons was performed to determine statistical differences among the formulations in their adhesive properties and 5-FU drug deposition in the skin samples. Three independent experiments were conducted to present results as the mean ± standard deviation (SD) unless otherwise indicated. Statistical significance was determined at *p* = 0.05.

## 3. Results and Discussion

### 3.1. Determination of Clinically Relevant 5-FU Dose in the Patches

An FDA-approved commercial product (Efudix^®^ cream, 5%) has been used for the treatment of AK, SCC in situ, and superficial BCC [23]. One of the significant clinical challenges in the application of Efudix^®^ cream for skin cancer treatment is the lack of consensus among clinicians regarding the optimal dosing. Moreover, accurate dosing information is not available in the product information [24]. This could potentially lead to inconsistent clinical outcomes as patients may apply either excessive or insufficient quantities of the cream. In this study, we aimed to derive a clinically relevant dose of 5-FU for NMSC by extrapolating criteria from dermatological literature. Utilising the concept of a fingertip unit suggests that 500 mg of topical formulations can cover the area of two adult hands with an estimated area of application of 320 cm^2^ [25]. Taking into account the twice-daily application of Efudix^®^ cream, a clinically relevant dose would be approximately 3 mg/cm^2^ (equivalent to 150 µg/cm^2^ of 5-FU content) over 24 h. Based on this estimation, the DIA patches were developed and optimised containing 450 µg/cm^2^, which can potentially be used over 72 h.

### 3.2. Slide Crystallisation Studies

The slide crystallisation studies were performed to determine the maximum loading of 5-FU that can be added to a EuE DIA matrix. Figure 1 demonstrates the absence of 5-FU crystals up to 5% (*w*/*w*, drug to polymer ratio) whilst increasing amounts of 5-FU crystals were observed as the 5-FU concentration increased beyond 6% in the polymer matrix. As the crystallisation of the active ingredient in the patches has significant implications for the efficacy and safety of the formulation, the maximum amount of 5-FU used throughout this study was kept at 5% to minimise the chances of 5-FU crystallisation [26]. On the other hand, the presence of 5-FU crystals in the samples could also be verified using additional methods, such as X-ray diffraction.

### 3.3. Preparation of 5-FU-Loaded Drug-in-Adhesive Patches

The DIA patches consist of three layers in which the drug-containing adhesive layer is placed between the release liner and backing layer (Figure 2A). The best combination of the backing layer and release liner was found to be Cotran™ 9720 and Scotchpak™ 9709, respectively, as determined in our previous study [3]. In this study, we aimed to develop and optimise 5-FU DIA patches containing a permeation enhancer, which can be applied over 72 h not only to achieve deeper skin penetration but also help to improve patient compliance.

Firstly, the thickness of DIA patches was optimised by screening three different wet casing thicknesses (500 µm, 750 µm, and 1000 µm) to contain the desired 5-FU content per cm^2^. The wet casting thickness of 500 µm, 750 µm, and 1000 µm resulted in the dried overall patch thickness of 266.7 ± 15.1 µm, 358.3 ± 7.5 µm, and 496.7 ± 17.5 µm, which contained 424.8 ± 24.2 µg/cm^2^, 766.2 ± 24.2 µg/cm^2^, and 1192.7 ± 34.9 µg/cm^2^ of 5-FU, respectively (Appendix A). To achieve the target loading of 450 µg/cm^2^, the wet thickness of 500 µm was used as a guide to prepare the patches containing different chemical permeation enhancers as per Table 1.

Three different FDA-approved chemical permeation enhancers for topical applications were selected from different classification categories. Based on their chemical properties, OA, Transcutol^®^, and Tween^®^ 20 can be classified as carboxylic acid, alcohol, and ester, respectively [17]. All prepared patches containing chemical permeation enhancers had a uniform surface without any signs of the crystallisation of 5-FU (Figure 2B), were highly flexible (folding endurance > 100), and had a uniform overall thickness (285.0 ± 8.4 µm to 290.0 ± 11.0 µm), weight (13.4 ± 0.6 mg/cm^2^ to 14.0 ± 1.1 mg/cm^2^), and drug content (462.7 ± 47.2 µg/cm^2^ to 487.1 ± 65.9 µg/cm^2^) (Table 2). These patches with permeation enhancers were found to be marginally thicker and heavier and contained a higher amount of 5-FU than the Patch-CONT by approximately 10%. It may be due to the inclusion of chemical permeation enhancers in the formulation as a less or non-volatile compound, which resulted in slightly thicker films after drying compared to the Patch-CONT. The surface pH of the patches ranged from 6.06 ± 0.03 to 6.33 ± 0.07 for Patch-CONT and Patch-OA, which were all found to be within the normal range of the skin, indicating that potential local irritation caused by the patches to the skin would be minimal [27]. Moreover, the % moisture content and % moisture absorption of the patches were also found to be minimal, ranging from 1.60 ± 0.35% to 2.33 ± 0.82% and 2.67 ± 0.85% to 3.91 ± 1.53%, respectively. These low values obtained by the patches practically help to prevent potential bacterial growth and to improve the long-term stability of the formulations, respectively [28].

### 3.4. Effect of Permeation Enhancers on Adhesive Properties of 5-FU-Loaded Drug-in-Adhesive Patches

Adhesive properties are one of the key quality attributes of topical patches to ensure their optimal performance [20]. The entire surface of DIA patches should come in close contact with the skin whereby the hydration of the skin can increase the partitioning of the drug from the patch to the skin surface for the initiation of permeation [29]. Tack is a surrogate measurement of the initial attachment of the DIA patches to the skin surface with gentle pressure. In this study, all of the patches demonstrated good initial bonding to the surface with the tack values ranging from 534.0 ± 108.3 g to 648.9 ± 123.1 g for Patch-CONT and Patch-TRAN, respectively (Figure 3A). The tack value of the Patch-CONT was also consistent with our previous study that involved the optimisation processes of the adhesive properties of EuE DIA patches [3]. Whilst there were a few other prior studies that developed EuE-based films, evaluating their suitability for topical patches was found to be challenging due to the absence of adhesion testing or the use of inconsistent testing methodologies between studies [30,31,32]. On the other hand, our previous study was the first to provide a comprehensive evaluation of the adhesive properties of EuE-based patches for topical applications using various in vitro models, benchmarking them against several commercially available products for topical applications [3]. In this study, the statistical analysis demonstrated that the incorporation of a chemical permeation enhancer (up to 10% of polymer quantity) into the patches did not significantly change the tack values.

On the other hand, the removal of the patches upon the completion of therapies should not cause excessive discomfort or pain and leave any formulation residues on the skin surface. The patches were cleanly removed from the stainless-steel surface without leaving any residues upon the completion of peel studies. The 90° peel adhesions ranged from 53.5 ± 2.5 g/mm to 57.7 ± 2.0 g/mm for Patch-OA and Patch-CONT with no statistical difference between the formulations (Figure 3B). Despite a wide acceptance of the stainless-steel surface for in vitro testing of adhesive properties due to its good data reproducibility, it does not fully simulate the physiological skin conditions, which may need to be complemented with in vivo human data [20,33].

### 3.5. Ex Vivo Permeation and Deposition Studies

Initial ex vivo deposition studies were performed over 24 h to determine the effect of different chemical permeation enhancers of the DIA patches on the skin deposition of 5-FU. Figure 4 shows various amounts of 5-FU quantified on the skin surface as well as across different skin layers. Interestingly, the amount of 5-FU found on the skin surface (as determined by the 5-FU content in skin wash) was significantly lower with the patches compared to Efudix^®^ cream. From the practical point of view, this may be attributed to the nature of the dosage form as patches would generally leave a very small amount of drug residues on the skin surface upon removal whilst any cream formulations would often require thorough washing to remove the excess drug from the skin surface. Moreover, the 5-FU content found in the SC was much higher from the cream formulation (62.78 ± 18.9 µg/cm^2^) compared to the Patch-CONT (24.47 ± 13.6 µg/cm^2^), Patch-TRAN (11.33 ± 3.1 µg/cm^2^) and Patch-OA (28.34 ± 12.5 µg/cm^2^). This may indicate that the 5-FU from the commercial formulation was not effectively partitioned out from the SC and penetration into the deeper tissues as a target site where more invasive NMSC grows [34,35]. Whilst the patch formulations did not show a statistically significant difference in epidermis and dermis deposition, Patch-TRAN (76.39 ± 27.7 µg/cm^2^) and Patch-OA (82.56 ± 8.2 µg/cm^2^) demonstrated much deeper penetration, as indicated in the 5-FU recovery from the receptor chamber than Efudix^®^ cream (34.63 ± 8.3 µg/cm^2^). As Patch-OA and Patch-TRAN were the best-performing formulations, the results were consistent with a previous study in which excised human abdominal skin pre-treated with OA demonstrated a 24-fold increase in the flux of 5-FU solution, compared to the control without any permeation enhancer [10]. The authors suggested that OA increased 5-FU permeation through the disruption of the intercellular lipid domain of the SC. Similarly, the enhanced permeation (1.23-fold) of 5-FU was also demonstrated in a solution containing Transcutol^®^ as a permeation enhancer applied on the excised mouse skin [36]. Transcutol^®^ is known for its unique ability to easily penetrate the SC and interact intensively with the water within the intercellular path. Such interactions can alter the barrier of the SC in multiple ways by adjusting the molecular mobility of the proteins and lipids to resemble that of skin soaked in the phosphate-buffered solution, effectively reducing the barrier function of the skin [37].

### 3.6. In Vitro Permeation Studies for 72 H

Considering the patches were designed for the 3-day application, in vitro permeation studies over 72 h using Strat-M^®^ membranes were conducted to evaluate their feasibility. Figure 5 illustrates that Patch-OA (31.07 ± 3.93 µg/cm^2^) and Patch-TRAN (33.05 ± 5.18 µg/cm^2^) delivered significantly more 5-FU whilst Patch-T20 (20.49 ± 6.12 µg/cm^2^), Patch-CONT (20.36 ± 2.92 µg/cm^2^), and Efudix^®^ cream (19.70 ± 3.77 µg/cm^2^) performed similarly within the first 24 h. All the patches exhibited continuous permeation over the 72 h with Patch-OA, Patch-Tran, Patch-T20, and Patch-CONT reaching 5-FU concentrations of 48.28 ± 3.51 µg/cm^2^, 49.20 ± 4.77 µg/cm^2^, 39.67 ± 3.80 µg/cm^2^, and 36.27 ± 5.28 µg/cm^2^, respectively. These findings align with the ex vivo study validating the superior skin permeation of 5-FU achieved by Patch-OA and Patch-TRAN, although the synthetic membranes were much less permeable than the excised pig ear skin. In addition, the permeation data over 72 h demonstrates their great potential as 3-day patches. On the other hand, permeation data for Efudix^®^ cream was not obtained beyond 24 h due to its clinical limitation of requiring application once or twice daily.

In this study, the main purpose of choosing Strat-M^®^ membranes was to strengthen the validity of our deposition and permeation data obtained previously ex vivo by providing complementary insights from both a biological and a synthetic model. The synthetic membranes allow for permeation data with high reproducibility, whilst permeation studies conducted for an extended period of up to 72 h using excised dermatomed skin could potentially result in high lot-to-lot variability, potentially affecting the validity of the outcomes [38,39]. The Strat-M^®^ membranes were developed to replicate the layered structure and lipid composition of human skin. Each Strat-M^®^ membrane has an approximate thickness of 300 µm and consists of a top layer supported by other two layers of porous polyether sulfone atop a single layer of polyolefin non-woven fabric support. The thickness and porosity of the membrane layers progressively increase to simulate the different layers of human skin, including the epidermis, dermis, and subcutaneous tissue [40]. Several permeation studies conducted on Strat-M^®^ membranes have demonstrated high correlations with human skin in the evaluation of various topical formulations containing amphotericin B [39], hydrocortisone [41], lidocaine [42], capsaicin [43], and several non-steroidal anti-inflammatory drugs [44] for various cutaneous diseases. In these studies, the in vitro permeation data were used as a surrogate measure for assessing drug penetration into deeper skin tissues, which is also crucial for effectively treating more aggressive forms of NMSC. However, this approach may also possess a potential risk of increased systemic exposure to 5-FU, which highlights a significant research gap in our understanding of the pharmacokinetics and biodistribution of topically applied 5-FU for balancing its efficacy and safety. Indeed, our study was the first to report the 5-FU deposition and permeation through in vitro and ex vivo settings from adhesive topical patches relevant to skin cancer. Future in vivo animal pharmacokinetic and cytotoxicity studies in conjunction with potential application of mathematical modelling, such as in vitro–in vivo correlation, would be required to better understand the distribution, metabolism, and potential toxic effects of these drugs within the body, thereby ensuring their efficacy and safety for topical NMSC treatment.

### 3.7. Stability Studies

The 5-FU DIA patches demonstrated excellent stability over 6 months when stored at room temperature (25 ± 2 °C, 65% RH) without any significant changes in the physical appearance, 5-FU contents (Figure 6A), and tack adhesion (Figure 6B).

### 3.8. In Vivo Evaluation of the Adhesive Properties and Safety of the Patches

As Patch-OA and Patch-TRAN were determined to be the best-performing formulations in vitro, their clinical feasibility as a 3-day patch was evaluated in vivo involving 11 healthy volunteers. Due to the known severe cytotoxicity of 5-FU, their drug-free counterparts were prepared using the same preparation method without the addition of 5-FU and were used throughout this in vivo study. Out of 13 volunteers initially enrolled in the study, a total of 11 completed and returned the questionnaire after completing the 72 h study (Appendix A). The detailed scores and key findings are summarised in Table 3. Overall, the participants reported both patches offered good adhesive and excellent wear comfort over 72 h with no statistically significant difference between them. Whilst no one reported local side effects of Patch-TRAN, two participants from Patch-OA experienced mild redness, blemishes, and swelling around the application area which subsided after 24 h without treatment. Such local skin irritations of OA had been reported previously in a small number of participants [45], which may be indicative of the safer nature of Patch-TRAN than Patch-OA. Indeed, the findings from our inaugural in vivo study of these topical patches demonstrate their potential as an innovative user-friendly long-acting drug delivery system towards NMSC topical treatment. The full individual participant’s responses can be found in the Appendix A.

## 4. Conclusions

In conclusion, our study effectively incorporated a therapeutically relevant high dose of 5-FU into EuE-based DIA patches, with Patch-OA and Patch-TRAN showing the highest skin permeability enhancement using both pig ear skin and Strat-M^®^ synthetic membranes. From the clinical perspective, these patches not only offer significant potential as a long-acting topical solution for up to three days with improved efficacy but also provide patients with enhanced compliance, as demonstrated in our in vivo human study. This study would facilitate further evaluation of the efficacy of the patches in various in vivo NMSC animal models underscoring the potential of these patches in addressing the widespread challenge of NMSC treatment. Moreover, the scalability of this patch technology and its adaptability for other therapeutic agents open avenues for broadening the scope of topical drug delivery systems. This research not only addresses the immediate challenge of topical NMSC treatment but also sets a precedent for future innovations in topical drug delivery, potentially transforming patient care in dermatology and beyond.

## Figures and Tables

**Figure 1 pharmaceutics-16-00379-f001:**
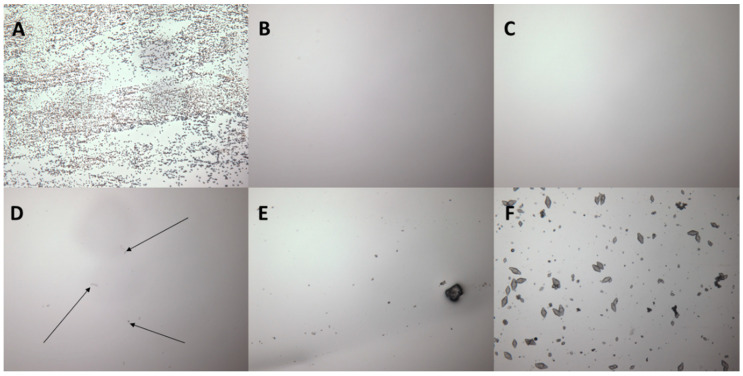
Images of slide crystallisation studies under an optical microscope showing increasing amounts of 5-FU crystals with an increase in 5-FU concentrations. (**A**) 5-FU crystals at 40×, (**B**) 3% (**C**) 5%, (**D**) 6%, with arrows indicating the presence of 5-FU crystals, (**E**) 7%, and (**F**) 10% at 10×.

**Figure 2 pharmaceutics-16-00379-f002:**
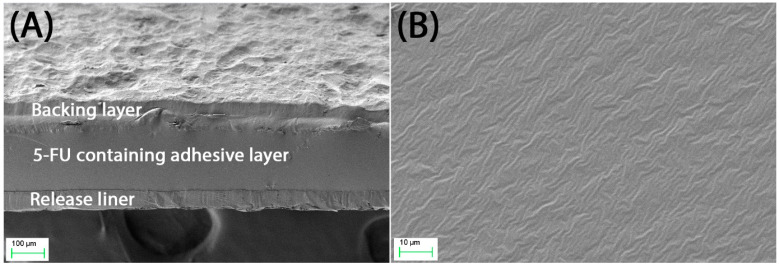
Scanning electron microscopy images showing (**A**) a cross-section of the 5-FU EuE patch and (**B**) the surface morphology of the 5-FU-containing adhesive layer.

**Figure 3 pharmaceutics-16-00379-f003:**
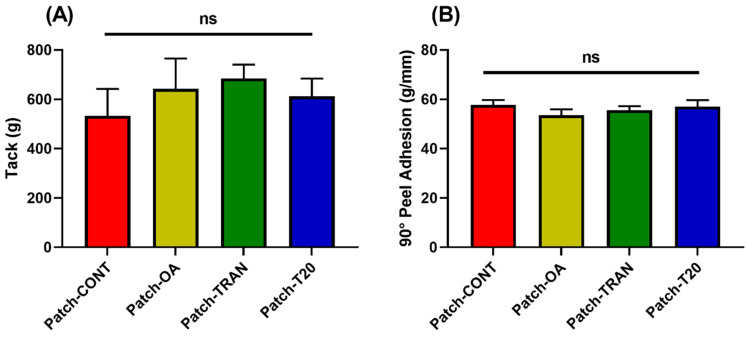
Assessment of adhesive properties of the patches containing different chemical permeation enhancers on (**A**) tack and (**B**) 90° peel adhesion. ns: Not statistically significant.

**Figure 4 pharmaceutics-16-00379-f004:**
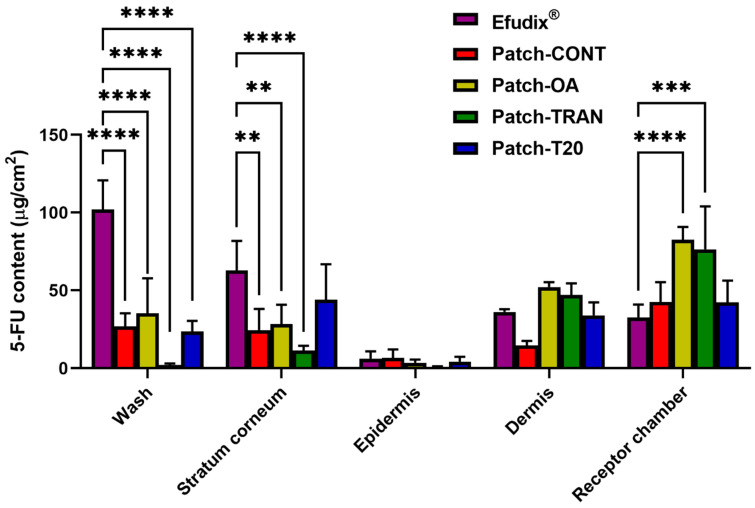
5-FU depositions in the different skin layers after 24 h. ****, ***, and ** indicate *p* < 0.0001, 0.001, and 0.01, respectively.

**Figure 5 pharmaceutics-16-00379-f005:**
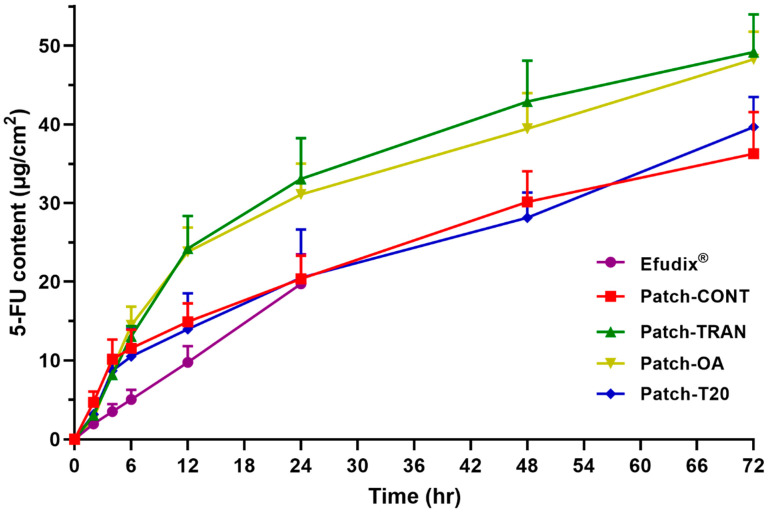
In vitro permeation studies over 72 h using Strat-M^®^ membranes.

**Figure 6 pharmaceutics-16-00379-f006:**
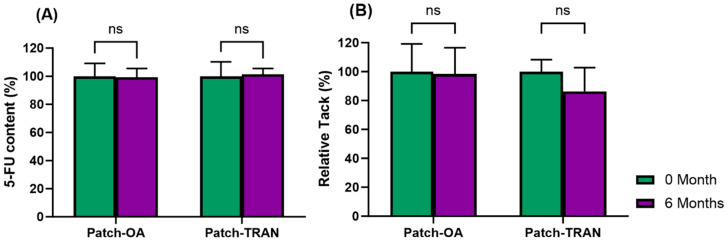
Stability studies of the 5-FU patches at room temperature (25 ± 2 °C, 65% RH) over 6 months. (**A**) 5-FU content in the patches and (**B**) tack adhesion. ns: Not statistically significant.

**Table 1 pharmaceutics-16-00379-t001:** Composition of 5-FU-loaded drug-in-adhesive mixtures with various chemical permeation enhancers.

Formulation	Ingredients							
	5-FU (g)	EuE (g)	Triacetin (g)	Succinic Acid (g)	Oleic Acid (g)	Transcutol^®^ (g)	Tween 20 (g)	MeOH (g)
Patch-CONT	0.4	8	3.2	0.4				To 20 g
Patch-OA					0.8			
Patch-TRAN						0.8		
Patch-T20							0.8	

**Table 2 pharmaceutics-16-00379-t002:** Thickness, weight, folding endurance, surface pH, % moisture content, % moisture absorption, and drug content of the prepared patches containing various chemical permeation enhancers.

Formulation	* Thickness (µm)	^ Weight (mg/cm^2^)	Folding Endurance (Times)	Surface pH	% Moisture Content	% Moisture Absorption	Drug Content (µg/cm^2^)
Patch-CONT	266.7 ± 15.1	12.4 ± 0.4	>100	6.06 ± 0.03	2.16 ± 0.65	3.42 ± 1.00	424.8 ± 24.2
Patch-OA	286.7 ± 19.7	13.9 ± 1.0	>100	6.33 ± 0.07	2.08 ± 0.99	3.38 ± 1.34	465.2 ± 42.6
Patch-TRAN	290.0 ± 11.0	13.4 ± 0.6	>100	6.21 ± 0.04	1.98 ± 0.55	2.67 ± 0.85	462.7 ± 47.2
Patch-T20	285.0 ± 10.5	14.0 ± 0.8	>100	6.17 ± 0.06	1.60 ± 0.35	3.61 ± 0.46	447.7 ± 35.7

* Overall thickness of the patches including the drug-in-adhesive layer, the release liner (75 µm) and the backing layer (76.2 µm). ^ Dry weight of the patches including the drug-in-adhesive layer and backing membrane.

**Table 3 pharmaceutics-16-00379-t003:** A summary of the in vivo human evaluation of drug-free DIA patches.

Question	Score		Key Findings
	Patch-OA	Patch-TRAN	
Q1. Ease of removal from the release liner	4.9 ± 0.3	4.9 ± 0.3	Most participants found the removal process of the patches from the release liner extremely easy; one participant reported it could potentially improve.
Q2. Initial tack	4.8 ± 0.4	4.8 ± 0.4	Most participants agreed that both patches provided excellent initial adhesion.
Q3. Ongoing adhesion for 72 h	4.8 ± 0.4	4.9 ± 0.3	Most participants reported suitable ongoing adhesion of the patches over 72 h on both arms; however, three participants reported that one Patch-OA accidentally fell off one arm after 48 to 60 h; one participant experienced the adhesion failure of one of Patch-TRAN at 60 h.
Q4. Wear comfort	5.0	5.0	All participants agreed that both patches were extremely comfortable to wear for 72 h.
Q5. Ease of removal from skin at 72 h	4.5 ± 0.8	4.5 ± 0.7	Three participants commented that both patches required a bit of effort to remove from the skin after 72 h.
Q6. Side effects	See key findings	Nil	One participant reported mild redness and swelling from Patch-OA around the application area; another participant experienced slight blemish surrounding the application area; symptoms subsided in 24 h; no side effects were reported from Patch-TRAN
Other comments	-	-	One participant commented that the transparency of the patches blended well with the skin, providing a good aesthetical look.

## Data Availability

The data presented in this study are available in this article and the Appendix A.

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
