# Peer review of "Enhanced Skin Permeation of 5-Fluorouracil through Drug-in-Adhesive Topical Patches"

_pharmaceutics, 2024, doi:10.3390/pharmaceutics16030379_

Round 1

Reviewer 1 Report

Comments and Suggestions for Authors

The paper investigates the topical administration of 5-FU for the treatment of NMSCs by applying skin patches as a vehicle.

The introduction of the work is well conducted, and the materials and methods are adequately described. The results are also very detailed.

The study also presents some tables and diagrams that enhance its understanding and provide a clear summary of the data presented.

Following some studies mentioned in the bibliography, the authors elaborate on the administration of the drug through skin patches that can make such administration easier, precise, comfortable, increase patient compliance, and above all release a correct amount of the active ingredient.

The principle of standardizing a skin application, such as a cream or solution, makes treatments more homogeneous and less subject to individual variations.

Conclusions are appropriate and consistent with the study.

English is fluent.

All in all, the paper is interesting especially from a pharmacokinetic perspective and opens the way for subsequent clinical studies.

Reviewer 2 Report

Comments and Suggestions for Authors

Dear Author,

Your manuscript titled "Enhanced Skin Permeation of 5-fluorouracil through Drug-in-Adhesive Topical Patches Towards Improved Non-Melanoma Skin Cancer Treatment" has undergone a thorough review. Following a meticulous assessment of its originality and content alignment, it has been concluded that your manuscript is in accordance with the criteria established by the Pharmaceutics journal. Minor issues have been found in your work. Fixing these can make your already good study even better. Congratulations on your work, and wish you continued success.

The minor issues are as follows:

1. Please rewrite the words from Latin origin. Such as in vitro, in vivo, in silico, Stratum corneum (the “s” should be capital letter).

2. Please clarify the release medium (was it containing sodium azide or it was used for only skin integrity evaluation.

3.  The in vitro release kinetics should be interpreted by using mathematical models (zero order, first order, Higuchi and/or Korsmeyer-Peppas…etc.) Then, the release mechanism could be illuminated. 

Respectfully,

Reviewer 3 Report

Comments and Suggestions for Authors

The manuscript pharmaceutics-2901665 “Enhanced Skin Permeation of 5-fluorouracil through Drug-in-Adhesive Topical Patches Towards Improved Non-Melanoma Skin Cancer Treatment” by Sangseo Kim et al. describes the development of a topical patch for improved dermal delivery of 5-fluorouracil. 

The manuscript is well written; the authors used modern research methods. The used references are reasonable.  

Questions and comments:

  1. The title of the manuscript should be revised by removing "Improved Non-Melanoma Skin Cancer Treatment" because clinical aspects are not presented in the paper. 

  2. Information on the biopharmaceutical benefits of patches (e.g., improved local bioavailability, programmed and controlled prolonged drug release, unidirectional drug release, reduced frequency of side effects, etc.) and the importance of their bioadhesive properties should be added to the Introduction. Some references may be helpful to you, for instance, doi.org/10.3390/ijms232112980 

  3. Section 2.6.3. - Ex vivo model should be described in more detail, information on the source of the skin should be added. Also, the composition of the medium should be specified.

  4. Section 3.2. - X-ray diffraction can be used to confirm the structure of 5-FU.

  5. The Conclusion should be written in detail to emphasize the great promise of your research.

Round 2

Reviewer 3 Report

Comments and Suggestions for Authors

The manuscript may be accepted